# SRPX Emerges as a Potential Tumor Marker in the Extracellular Vesicles of Glioblastoma

**DOI:** 10.3390/cancers14081984

**Published:** 2022-04-14

**Authors:** Elisabet Ampudia-Mesias, Samia El-Hadad, Charles Scott Cameron, Adelheid Wöhrer, Thomas Ströbel, Nurten Saydam, Okay Saydam

**Affiliations:** 1Department of Pediatrics, Division of Hematology and Oncology, Medical School, University of Minnesota, Minneapolis, MN 55454, USA; ampud001@umn.edu (E.A.-M.); camer308@umn.edu (C.S.C.); 2Department of Reproductive Endocrinology, University Hospital Zurich, 8006 Zurich, Switzerland; samia.el-hadad@usz.ch; 3Division of Neuropathology and Neurochemistry, Department of Neurology, Medical University of Vienna, A-1090 Vienna, Austria; adelheid.woehrer@meduniwien.ac.at (A.W.); thomas.stroebel@meduniwien.ac.at (T.S.); 4Department of Biochemistry, Molecular Biology, and Biophysics, Medical School, University of Minnesota, Minneapolis, MN 55454, USA; nurtensaydam@yahoo.com

**Keywords:** SRPX, EVs, proteomics, glioblastoma

## Abstract

**Simple Summary:**

Glioblastoma is the most common malignant primary brain tumor and remains incurable. Additionally, there are only a few non-invasive early diagnostic and prognostic markers for this disease. The stability of extracellular vesicles (EVs) and their availability in patient serum make them ideal for discovery of early markers associated with diagnosis, prognosis, and treatment response for glioblastoma. In this study, we used proteomics analysis to discover a novel tumor biomarker in glioblastoma human primary cell-derived EVs and found that sushi-repeat containing protein X-linked (SRPX) was the only protein identified in the majority of glioblastoma EVs that was absent in the HPA-derived EVs. Moreover, we further analyzed the possible role of SRPX in glioblastoma tumorigenesis and found that SRPX is involved in glioblastoma cell growth, and SRPX depletion sensitizes glioblastoma to temozolomide (TMZ). Taken together, our results suggest that SRPX can be used as a novel tumor biomarker for diagnostic and prognostic purposes for glioblastomas.

**Abstract:**

Extracellular vesicles (EVs) may be used as a non-invasive screening platform to discover markers associated with early diagnosis, prognosis, and treatment response. Such an approach is invaluable for diseases such as glioblastoma, for which only a few non-invasive diagnostic or prognostic markers are available. We used mass spectrometry to analyze proteomics profiles of EVs derived from four glioblastoma cell lines and human primary astrocytes (HPAs) and found that SRPX is the only protein enriched in the majority of glioblastoma EVs that was absent in the HPA-derived EVs. Then, we evaluated the relationship between SRPX protein expression and tumor grade using immunohistochemical staining (IHC) and performed colony formation and viability assays to analyze the possible function of SRPX in glioblastoma. SRPX mRNA and protein expression were associated with tumor grade. Moreover, temozolomide (TMZ)-resistant tumor tissues showed highly positive SRPX staining, compared to all other tumor grades. Additionally, glioblastoma cells displayed enhanced SRPX gene expression when exposed to TMZ. Knockdown of SRPX gene expression via siRNA inhibited cell viability. Taken together, the results of this study suggest that SRPX can be used as a novel tumor marker for diagnostic and prognostic purposes and can also be a therapeutic target for glioblastomas.

## 1. Introduction

Glioblastomas are highly malignant tumors that account for almost 50% of all high-grade gliomas [1]. Despite combined use of debulking surgery, radiotherapy, and temozolomide (TMZ), the median survival remains less than two years [2]. Current efforts are directed towards the identification of tumor biomarkers that facilitate cancer screening and detection, disease monitoring, predicting prognosis, and survival after clinical intervention.

Extracellular vesicles (EVs) refer to a diverse group of vesicles that differ with respect to their origin, size, and biological content [3]. These vesicles are secreted by almost all cell types and play key roles in mediating intercellular communication. The following two features of EVs make them promising tools to identify markers associated with diseases: the presence of bioactive molecules -mRNAs [4], microRNAs (miRNA) [5], proteins and DNA [6], and their membrane composition, which makes them stable carriers [7,8]. The discovery of EVs in the blood of cancer patients provides a novel type of biomarker for various patient scenarios [9,10]. This was reflected in recent studies, which have pointed out the possible use of miRNA and mRNA content of EVs as a novel cancer diagnosis tool [4,11,12].

Proteomics is a popular high-throughput screening approach to identify disease associated proteins. Many proteomics studies on glioblastoma have relied on established cell lines [13,14], tumor tissues, plasma [15], and cerebrospinal fluid (CSF) [16]. A recent study explored how EGFRvIII activation affects the proteome of glioblastoma-derived EVs [17].

Here, we used four patient-derived primary glioblastoma cell lines and human primary astrocytes (HPAs) to identify differentially expressed proteins in EVs, which may serve as putative disease markers for glioblastoma. We identified that SRPX was the only protein enriched in the majority of glioblastoma cell line EVs (three out of four cell lines EVs) that was absent in the HPA-derived EVs. To validate this finding, we performed immunohistochemistry and RT-qPCR experiments on Grade 2–4 gliomas from tumor tissue samples, as well as in silico data analysis using R2: Genomics Analysis and Visualization Platform (https://hgserver1.amc.nl/cgi-bin/r2/main.cgi, accessed on 12 January 2022).

## 2. Materials and Methods

### 2.1. Cell Lines

Glioblastoma cell lines IN-GB-11, IN-GB-28, IN-GB-29, and IN-GB-9 were established from fresh tumor biopsies according to standard cell culture techniques [18] at the Institute of Neurology, Medical University of Vienna. All typical mutations (e.g., TP53, PTEN) present in the cell lines were confirmed by mutation analysis of the corresponding tumor tissue. Human primary astrocytes (HPAs) were purchased from Applied Biological Materials (ABM) Inc. (ABM Inc., Richmond, BC, Canada). HPAs were cultured on collagen-coated dishes in Prigrow IV medium (ABM Inc., Richmond, BC, Canada), supplemented with 10% fetal bovine serum (FBS; Gibco, Grand Island, NY, USA). Human glioblastoma cell lines U87-MG, U251-MG, and human embryonic kidney 293T cells (HEK293T) were purchased from the American Type Culture Collection (ATCC; Manassas, VA, USA). U251-MG and HEK293T cells were cultured in Dulbecco’s modified Eagle medium (DMEM; Gibco, Grand Island, NY, USA), and U87-MG cells were cultured in Minimum Essential Media (MEM; Gibco, Grand Island, NY, USA). Both media, DMEM and MEM, were supplemented with 10% FBS (FBS; Gibco, Grand Island, NY, USA), 100 mg/mL penicillin G, and 100 lg/mL streptomycin (Gibco, Grand Island, NY, USA). All cell lines were cultured at 37 °C in a humidified environment with 5% CO_2_.

### 2.2. Extracellular Vesicle (EV) Isolation

Glioblastoma cell lines were plated at 5 × 10^6^ cells per 10 cm dish. The next day, culture media was replaced with 5% exosome-depleted FBS (ExoFBS, System Biosciences, Palo Alto, CA, USA). Culture media were collected 72 h later, and EVs were isolated with Total Exosome Isolation Reagent from cell culture media (Thermo Fisher Scientific, Waltham, MA, USA) according to the manufacturer’s instructions. EVs were resuspended in PBS and stored at −80 °C.

### 2.3. Nanoparticle Tracking Analysis (NTA)

EV samples were analyzed on a NanoSight NS500 instrument (Malvern Panalytical, Amesbury, UK), as described previously [19]. Each sample was measured six times, and the average size and average total particle number were calculated.

### 2.4. Protein Isolation from EVs

EV samples were lysed in a 5× lysis buffer (Tris 1M pH 8.0, 0.5 M NaCl, glycerin, 10% NP-40, 0.5 M EDTA, protease inhibitor medley). Samples were incubated on ice for 30 min, and then centrifuged at 13,800× *g* for 30 min.

### 2.5. Mass Spectrometry

Proteomics experiments were carried out in the Proteomics Core Facility at the Medical University of Vienna. Prior to digestion, proteins were precipitated using a methanol–chloroform procedure, as described by Wessel and Flügge [20]. Precipitated proteins were dissolved in 50 mM of triethylammonium bicarbonate and the protein concentration was determined using the Bradford assay. Proteins were digested with trypsin overnight at 37 °C, using a trypsin/protein ratio of 1/50. Digestion was stopped by acidification with 10 µL of 1% trifluoroacetic acid (TFA). For injection onto the separation column, 20 µL of digested peptides were further diluted with 30 µL 0.1% TFA. Peptides were separated using an UltiMate Plus nano HPLC (LC Packings, Amsterdam, The Netherlands) separation system, which consists of a Famos autosampler, Switchos column switching unit, the UltiMate nano pump, and a UV detector. The Acclaim C18 trap column (300 µm ID × 5 mm) was operated at ambient temperature, and the Acclaim C18 nano separation column (75 µm ID × 250 mm) was mounted in the column oven and operated at 45 °C. Samples were loaded onto the trap column using 0.1% TFA at 30 µL/min, and nanoseparation was performed using a ternary gradient composed at 300 nl/min from the following: (a) 0.1 formic acid (FA) in 5% aqueous acetonitrile (AcN); (b) 0.08% FA in 15% methanol (MeOH), 15% AcN, 70% water; and (c) 0.08% FA in 60% AcN, 30% MeOH, and 10% 2,2,2-trifluoroethanol (TFE). A user defined injection program (UDP) was used for sample injection and additional injector and trap column wash. Every sample injection was followed by two blank runs with injections of TFE for removal of possible sample remains in injector or on the trap column, and prevention of carry-over in the separation system [21]. Mass spectrometric (MS) analysis was performed on the LTQVelos IonTrap mass spectrometer (Thermo Fisher, Bremen, Germany) with the “Top 20” method, in which the 20 most intensive ions from the MS scan were selected for MS/MS. Single charged ions were excluded from fragmentation and detected ions were excluded for further fragmentation for three minutes after initial MS/MS fragmentation had been performed. In addition, UV peptide detection at 214 nm was also performed prior to MS. Data analysis was performed using Mascot 2.4.1 (Matrix Science, London, UK) for searching the most recent version of the SwissProt database and using the mass tolerance of 0.4 Da for MS and MS/MS. Identifications with two peptides per protein and a Mascot score of >40 were accepted. All search results were refined and researched using Scaffold 5.0.1 software with a minimum peptide threshold of 95%, minimum protein threshold of 99%, peptide FDR of 1.12%, and protein FDR of 0.0%. UniProt’s Retrieve/ID Mapping tool (Accessible at: https://www.uniprot.org/uploadlists/, accessed on 1 December 2019) was used to convert UniProt entry names to gene IDs. The summary statistics are provided in Appendix A. 

### 2.6. Immunohistochemical Staining

Three formalin-fixed, paraffin-embedded (FFPE) sections (5 μm thickness) from gliomas Grade 2, Grade 3, Grade 4 (glioblastoma), and TMZ-resistant tumors were used for immunohistochemical staining experiments. Relevant information about treatment received by the patients is presented in Appendix A. White matter was used as a negative control, and tonsil biopsy was used as a positive control. Sections were stained with an SRPX antibody (NBP1-77086, 1:500 dilution, Novus Biologicals, Centennial CO, USA). The EnVision K5007 system (Dako, Glostrup, Denmark) was used for secondary staining [18]. A digital camera was used to capture images of stained sections.

### 2.7. RNA isolation and RT-qPCR

Total RNA from glioblastoma, and HEK293T cell lines was isolated using Trizol (Ambion, Life Technologies, Thermo Fisher Scientific, Waltham, MA, USA). For glioblastoma cell lines, cDNA was synthesized with the QUANTAbio kit (Beverly, MA, USA), using 1000 ng of total RNA. cDNA was analyzed by quantitative PCR with the SYBR Green qPCR Mix, QUANTAbio System (Beverly, MA, USA). For each mRNA analyzed, three technical replicates per sample were used. A no-template control was included in each experiment to rule out contamination. A set of primer sequences was used to amplify SRPX (Appendix A), and the following primer sequences were used to amplify GAPDH or B-actin (Appendix A). The input cDNA amount was 15 ng per reaction.

### 2.8. Analysis of Chinese Glioblastoma Genome Atlas (CGGA) Datasets

Expression data (STAR + RSEM) of the mRNAseq_693 were downloaded from the CGGA website. Expression data was preprocessed and normalized with the edgeR package (v.3.32.1) [22].

### 2.9. Generation of Temozolomide Resistant U251-MG Cells

TMZ-resistant cells were generated as described in Ströbel et al. 2017 [18]. Briefly, U251-MG cells were exposed to increasing concentrations of TMZ (10–320 µM) (MilliporeSigma, Burlington, MA, USA) over a period of three months. One clone propagating in 320 µM of TMZ was isolated and cultured in the presence of TMZ for at least a month prior to the study.

### 2.10. Transfection

Transfection was performed by Lipofectamine RNAiMAX (Thermo Fisher Scientific, Waltham, MA, USA) according to the manufacturer’s procedure. 5 × 10^4^ cells/well were seeded in a 96 well plate and, after 24 h, cells were transfected with two different SRPX siRNAs or control or siRNA (Thermo Fisher Scientific, Waltham, MA, USA) at a final concentration of 60 nM or 100 nM. To assess knockdown efficiency, a separate set of transfections was performed and, seventy-two hours after transfection, total RNA was isolated from the transfected cells and RT-qPCR was performed to assess SRPX mRNA expression. 

#### 2.10.1. Cell Viability Assay

Cells were transfected using the same conditions described above and, 72 h after transfection, U251-MG-P, U251-MG-R, U87-MG, and HEK293T cells were stained with 0.4% trypan blue (Thermo Fisher Scientific, Waltham, MA, USA) and counted using a Neubauer cell counting chamber (MilliporeSigma, Burlington, MA, USA).

#### 2.10.2. Cell Viability Assay for U251-MG-R Cells Treated with TMZ

In a similar experiment, U251-MG-R were seeded at a density of 4 × 10^4^ cells/well in a 24 well plate and, after 24 h, cells were transfected with indicated siRNAs, SRPX siRNAs, or a control (Thermo Fisher Scientific, Waltham, MA, USA) at a final concentration of 60 nM. After 24 h, cells were exposed to 200 uM of TMZ and incubated for another 72 h. Then cells were stained with 0.4% trypan blue and counted using a Neubauer cell counting chamber (MilliporeSigma, Burlington, MA, USA).

### 2.11. Clonogenic Survival Assay

Cells were transfected with indicated siRNAs and, 24 h after transfections, U251-MG-P cells were incubated at different TMZ concentrations (10 µM, 25 µM, 50 µM) for 1 h. Cells were then seeded (500 cells/35 mm plates), and cultured for 10 days, followed by staining with 0.5% crystal violet solution (Sigma-Aldrich, St. Louis, MO, USA). Surviving colonies consisting of more than 50 cells were counted, and the percentage of survival was expressed as a ratio of survival against control cells treated with DMSO (MilliporeSigma, Burlington, MA, USA). The colonies were then fixed, stained with 0.5% crystal violet (Sigma-Aldrich, St. Louis, MO, USA), and visualized under a light microscope (AZ100M, Nikon Instruments Inc. Melville, NY, USA) (1× magnification).

### 2.12. TMZ Sensitivity Assay

Cells were seeded into a 96-well plate at a concentration of 5 × 10^4^. After 24 h, cells were exposed to increasing TMZ concentrations (10 µM, 25 µM, 50 µM, 75 µM, 100 µM, 200 µM, 300 µM) and incubated for 72 h. Cells were harvested and RNA was isolated after 72 h. In a parallel experiment, cells were seeded in a 96-well plate, exposed to 200 µM TMZ and incubated for different time intervals (1 h, 6 h, 12 h, 18 h, 24 h, 48 h, 72 h) to determine the SRPX gene expression.

### 2.13. Statistical Analysis

NTA measurements were expressed as mean ± standard deviation (SD). One-way ANOVA with post-hoc Tukey tests were used to compare SRPX mRNA expression among more than two groups in CGGA data sets. Dunn’s test was used as a post-hoc test to adjust for multiple comparisons. *p* values < 0.05 were considered as statistically significant. For qPCR, relative expression analysis was performed using the ΔΔCq method and gene expression was normalized to hB-actin or hGAPDH. Gene expression graphs were constructed using GraphPad Prism. Error bars indicate means ± SEM and statistical significance was compared to the control siRNA. *p* ≤ 0.05, *p* ≤ 0.01, *p* ≤ 0.001, *p* ≤ 0.0001.

## 3. Results

### 3.1. Proteomics Analysis of Glioblastoma- and HPA-Derived EVs

Nanoparticle Tracking Analysis of EVs from primary glioblastoma cell lines (IN-GB-9, IN-GB-11, IN-GB-28, IN-GB-29) and HPAs revealed that EVs had a mean size of approximately 90 nm (Appendix A).

To identify potential glioblastoma-specific markers, we used mass spectrometry to analyze EVs and cell lysates of four primary glioblastoma cell lines and HPAs (Figure 1A). We identified 10,799 spectra corresponding to 573 proteins (Appendix A). EV samples and corresponding cell lysates showed a clear separation on the basis of the overall proteomics similarity (Figure 1B). Among 573 proteins (Appendix A), 92 were specific to EVs (Figure 1C), which included some of the top 100 proteins frequently identified in exosomes [23] (Appendix A).

Strikingly, sushi repeat-containing protein, X-linked (SRPX) was the only protein detected in the majority of glioblastoma-derived EVs (three out of four cell line EVs) that was absent in the HPA-derived EVs. Therefore, in the present study, we focused on SRPX among the top dysregulated proteins. Previous efforts to characterize the proteome of EVs released from glioblastoma cells [17], as well as 60 human cancer cell lines (NCI-60) [24], revealed the presence of the SRPX protein in cancer cell EVs. In light of these observations and our proteomics data, we hypothesized that SRPX may be a disease marker for glioblastomas. To test our hypothesis, we performed immunohistochemical staining on FFPE tissue sections from gliomas Grade 2 (*n* = 3), Grade 3 (*n* = 3), Grade 4 primary (*n* = 3), and recurrent tumors (*n* = 3). SRPX protein expression was lowest in Grade 2 tumors and increased with tumor grade. Notably, the highest SRPX expression was detected in the TMZ-resistant recurrent tumors (Figure 2A). To validate these observations in larger and independent patient cohorts, we analyzed two expression datasets from the Chinese Glioblastoma Genome Atlas (CGGA), which contain RNA-seq and associated clinical data from 325 (mRNAseq_325) and 693 (mRNAseq_693) patients [25,26]. In the smaller dataset, which includes all gliomas, SRPX mRNA expression was higher in Grade 4 samples, but the differences among the groups were not statistically significant (Appendix A). Our result is congruent with the Human Protein Atlas database that includes all gliomas, and shows no statistical significance among gliomas (https://www.proteinatlas.org/ENSG00000101955-SRPX/pathology/glioma, accessed on 11 February 2022).

On the contrary, in the larger dataset, Grade 4 tumor samples had higher SRPX mRNA expression compared to Grade 3 samples (Figure 2B). Thus, our results suggest that SRPX is a marker for glioblastoma. Furthermore, we examined the SRPX expression in glioblastoma tumor tissues from the R2 database (http://r2.amc.nl, accessed on 12 January 2022). We found that SRPX is mainly expressed in glioblastoma tumors, when compared to control normal brain tissues (Figure 3). In conclusion, these results show that SRPX is expressed at higher levels in glioblastoma tumors, suggesting that SRPX might play a crucial role in glioblastoma tumorigenesis.

### 3.2. Association of SRPX with TMZ Resistance in Glioblastomas

SRPX enriched in EVs has been suggested to have a crucial role in matrix and membrane remodeling in drug resistance, in head and neck squamous cell carcinoma [27]. Moreover, SRPX has been shown to be a cancer stem cell marker related to chemoresistance in liver cancer [28]. In the light of these observations and based on our IHC results showing high SRPX protein expression in human glioblastoma tumors, (Figure 2A), we wanted to further evaluate the possible role of SRPX in glioblastoma TMZ resistance using different approaches. First, we evaluated the endogenous transcript expression levels of the SRPX by qPCR assay in two cell lines, U251-MG-P (parental) and U251-MG-R (TMZ resistant cells), and we found that SRPX was expressed at higher levels in U251-MG-R cells than in U251-MG-P cells (Appendix A). Next, we performed a dose-dependent assay using these cell lines. U251-MG-P or U251-MG-R cells were treated with increasing concentrations of TMZ (10 µM, 25 µM, 50 µM, 75 µM, 100 µM, 200 µM, 300 µM). Total RNA was isolated, and qPCR was performed 72 h after TMZ treatment. The data showed that the TMZ treatment resulted in significant upregulation of SRPX mRNA expression in both cell lines (Figure 4A), compared to cells treated with control DMSO. In U251-MG-P cells, SRPX transcript levels were significantly higher at 25 µM (*p* < 0.0001) and 50 µM (*p* < 0.0001) of TMZ treatment compared to control DMSO treatment cells, but dramatically dropped to the lower levels at 200 µM and 300 µM TMZ due to cell death (Figure 4A). In U251-MG-R cells, SRPX gene expression was significantly elevated at 100 µM TMZ (*p* < 0.0001; Fold change: 5.16), and at 200 µM (*p* < 0.0001; Fold change: 4.69) compared to the control treated cells. We next address the question of whether SRPX status upon TMZ treatment is time dependent. In order to address this question, we treated both U251-MG-P and U251-MG-R cells with 200 µM TMZ and incubated them for different time intervals (1 h, 6 h, 12 h, 18 h, 24 h, 48 h, 72 h), and found that SRPX gene expression was significantly increased at 6 h (*p* < 0.000001; Fold change: 3.50) and at 72 h (*p* < 0.000001; Fold change: 3.19) after TMZ treatment in U251-MG-R cells (Figure 4B), whereas there was no significant increase in U251-MG-P, as expected because these cells are not viable in this TMZ concentration.

### 3.3. SRPX Is Involved in Cell Viability of Glioblastoma Cells

We next asked a question about whether SRPX plays a role in cell growth of glioblastoma cells, and performed several experiments using glioblastoma cells and HEK293T cells as controls. Glioblastomas cells were seeded in a 96-well plate, transfected with the indicated siRNAs (Appendix A) and counted after 72 h. SRPX depletion dramatically inhibited the number of live cells in both U251-MG-P (~70% non-viable) and U251-MG-R cell lines (~75% non-viable), while the magnitude of reduction was smaller (~30% non-viable) in U87-MG cells (Figure 4C). However, the reduction in HEK 293T cells was considerably smaller following siRNA treatment (~15% non-viable), and not significantly different from control siRNA treated or untreated controls, suggesting a specific role of SRPX in glioblastoma tumorigenesis. Taken together, these results show that SRPX reduces cell viability in glioblastoma cell lines.

### 3.4. SRPX Depletion via siRNA Sensitizes Glioblastoma Cells to TMZ

In order to investigate whether SRPX is involved in TMZ response or resistance, we carried out a colony formation assay. Cells were transfected with indicated siRNA and, 24 h later, U251-MG-P and U251-MG-R cells were exposed to TMZ for 1 h. Afterwards, 500 cells were seeded in a 35 mm plate, cultured for 10 days, and surviving colonies were counted, fixed, stained with crystal violet and visualized under a microscope. We found that SRPX depletion significantly sensitized U251-MG-P cells to TMZ, compared to control siRNA treated and untreated cells (Figure 4D). We also observed significantly fewer numbers of colonies (Appendix A). In U251-MG R cells, we observed a decreased number of colonies and a clear trend in decreasing colony numbers between treatments, although there was no statistical significance (Appendix A). We next performed a cell viability assay, in order to investigate whether SRPX knockdown sensitizes resistant glioblastoma cells to TMZ. We transfected U251-MG-R cells with siRNA directed against SRPX mRNA and, 24 h later, treated with TMZ. Two days after treatment, we counted the surviving cells and found that the absence of SRPX sensitized U251-MG-R cells to TMZ (Appendix A). Taken together, our results show that SRPX depletion sensitizes glioblastoma cell lines to TMZ.

## 4. Discussion

In this study, our goal was to use proteomics to identify differentially expressed proteins in glioblastoma-derived EVs. Overall, we identified 92 proteins in glioblastoma-derived and HPA-derived EVs. We identified the following five proteins that were present in all samples, both glioblastoma-derived EVs and HPA-derived EVs: THBS1, FN1, TGFBI, ACTN4, and LGALS3BP. TGFBI is a collagen-interacting protein. TGFBI is expressed at higher levels in mesenchymal glioblastoma tumors [29,30] and is reported as a potential signature gene for the mesenchymal subtype [31]. LGALS3BP is regarded as a glioblastoma stem cell-associated marker, and a CD9 ligand. A study aiming to characterize the secretome of glioblastoma-derived neural stem cells (GNS) identified LGALS3BP protein to be upregulated in GNS cells [32]. Earlier studies to identify differentially expressed genes in glioma revealed that FN1 is overexpressed in gliomas, compared to normal brain tissue [33]. THBS1 plays a crucial role in tumor cells invasion in glioblastoma, whereas THBS1 is a gene with increased expression in high-grade gliomas compared to low-grade gliomas, its expression is confined to tumor cells and vessels, and the TGFβ1-induced THBS1 expression via Smad3 contributes to the invasive behavior during glioblastoma expansion [34]. SRPX, in contrast, was present in most of the glioblastoma-derived EVs and was absent in HPA-derived EVs, therefore, in this study we focused on SRPX and investigated its possible role as a tumor biomarker in glioblastoma tumorigenesis. 

There are a limited number of studies examining the role of SRPX in disease. SRPX is a transmembrane protein consisting of 464 amino acids and three sushi domains. It is also known as SRPX1 [35], ETX1 [36] and DRS [37]. A pioneering study on SRPX revealed SRPX deletion in patients with X-linked retinitis pigmentosa [38,39], and a recent study revealed that SRPX is a novel disease-associated molecule in cerebral amyloid angiopathy [35]. The role of SRPX in cancer was first reported on v-src (drs) studies, the mouse homolog of SRPX, which implied a tumor-suppressor role for this gene [40,41]. Supporting evidence for these observations came from expression profiling studies that reported downregulation of SRPX expression in multiple cancers [42,43,44], and subsequent studies suggested that SRPX plays a role in apoptosis [37] and senescence [45]. Additionally, recent studies pointed out a critical role of SRPX in cell migration and invasion of ovarian cancer cells. Knocking down of SRPX via shRNA technology resulted in significant inhibition of ovarian cancer cell migration. Notably, SRPX is among the soluble factors secreted by core-like glioblastoma cells and is assumed to contribute to intercellular signaling between core- and edge-like glioblastoma cells [46].

To validate the proteomics findings, we performed immunohistochemical staining on glioma Grade 2–4 tumors, which revealed that glioblastoma and TMZ-resistant recurrent tumors had the highest SRPX protein levels. Supporting evidence for these observations came from independent data analysis of published RNA-seq datasets, which contained significantly larger patient cohorts, showing that SRPX is significantly upregulated in glioblastomas compared to normal brain tissues and lower grade gliomas (Figure 2 and Figure 3) and suggesting that SRPX serves as a tumor marker only in glioblastomas. 

We next addressed the question of the possible function of SRPX upregulation in glioblastomas and found that siRNA depletion of the SRPX mRNA significantly reduced the cell viability and colony formation of glioblastoma cells, whereas SRPX depletion did not affect the cell viability of HEK293T cells that served as control, suggesting a specific role of SRPX in glioblastoma tumorigenesis. 

The link between SRPX and chemoresistance has not been explored in detail. A recent study showed that breast cancer cell lines resistant to trastuzumab emtansine have higher SRPX mRNA levels compared to naïve cells [47]. Our observation of high SRPX staining in TMZ resistant tumor samples triggered us to further evaluate the possible role(s) of SRPX in TMZ metabolism in glioblastomas. We first compared the endogenous transcript expression levels of the SRPX by qPCR assay in two cell lines, U251-MG-P (parental) and U251-MG-R (TMZ resistant cells), and found that SRPX was significantly overexpressed in U251-MG-R cells compared to the parental cells, U251-MG-P. We also found that TMZ treatment of glioblastoma cells showed a significant increase in the SRPX mRNA levels in a dose-dependent manner. Moreover, SRPX depletion via siRNA sensitizes glioblastoma cell lines to TMZ. Taken together, these findings suggest that SPRX is involved in TMZ metabolism, and further studies are required to describe the exact mechanism by which SRPX regulates TMZ metabolism in glioblastomas. 

This study has several important implications, although with some limitations. First, the true cell of origin for glioblastoma is a controversial subject. Experimental evidence points to at least three major cell types—neural stem cells (NSCs) including radial glial cells, oligodendrocyte precursor cells (OPCs), and astrocytes—that contribute to glioblastoma tumorigenesis [48,49,50]. Therefore, generalization of our findings may require validation in other cell types. Second, patient-derived cell lines may have acquired genetic changes over time, which are likely to be reflected in their proteome as well. To our knowledge, this is the first line of evidence that SRPX is enriched in EVs of glioblastoma cell lines and can serve as a novel tumor marker. However, further studies are required to investigate whether SRPX can be detected in patient serum-derived EVs as a circulating tumor marker. 

## 5. Conclusions

Taken together, our findings demonstrated that SRPX is highly enriched in glioblastoma EVs and provided insight into the clinical relevance of how this unique tumor marker could contribute to the early detection of glioblastomas. Furthermore, our study highlights the role of SRPX in glioblastoma tumorigenesis. Further studies are required to evaluate and validate in a larger cohort whether SRPX can be a diagnostic, prognostic and/or predictive marker to treatment response, in particular for TMZ in patient serum-derived EVs.

## Figures and Tables

**Figure 1 cancers-14-01984-f001:**
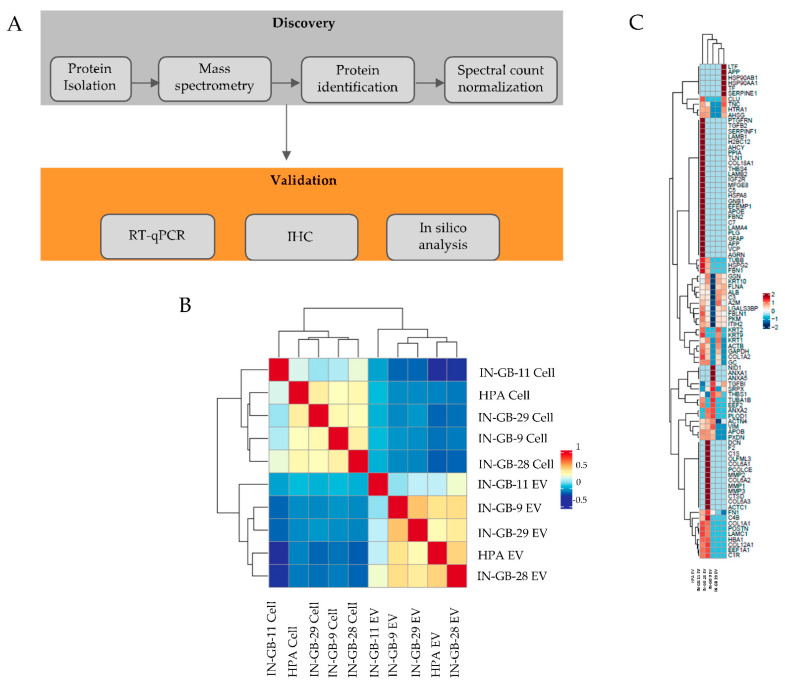
Experimental design. (**A**) Schematic depiction of experimental strategy to discover and validate glioblastoma-specific tumors; (**B**) Correlation of protein expression between EVs and their parental cell lines; (**C**) Heatmap showing the 92 proteins specific to EVs.

**Figure 2 cancers-14-01984-f002:**
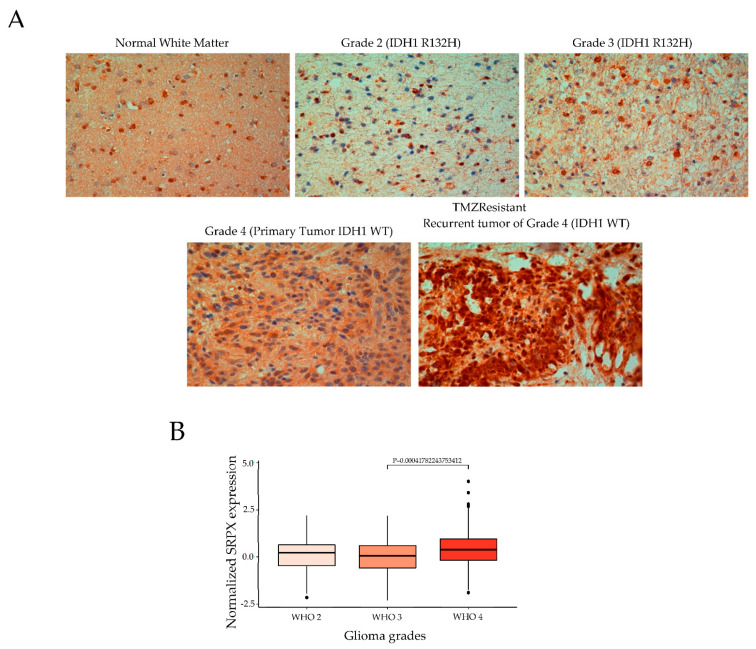
SRPX is a marker for glioblastomas. (**A**) Immunohistochemistry (IHC) of SRPX on FFPE tissue sections from grade 2 (*n* = 3), grade 3 (*n* = 3), grade 4 (*n* = 3) and recurrent tumors of grade 4 (*n* = 3) (lower panels show primary and recurrent tumor from the same grade patient). White matter biopsies were used as a negative control. Representative images from each grade are shown. Magnification 400×; (**B**) CGGA data analysis: normalized SRPX expression in glioma tumor grades in the mRNAseq_693 dataset. Kruskal-Wallis was used to compare SRPX mRNA expression among more than two groups and Dunn’s test was used to adjust for multiple comparisons.

**Figure 3 cancers-14-01984-f003:**
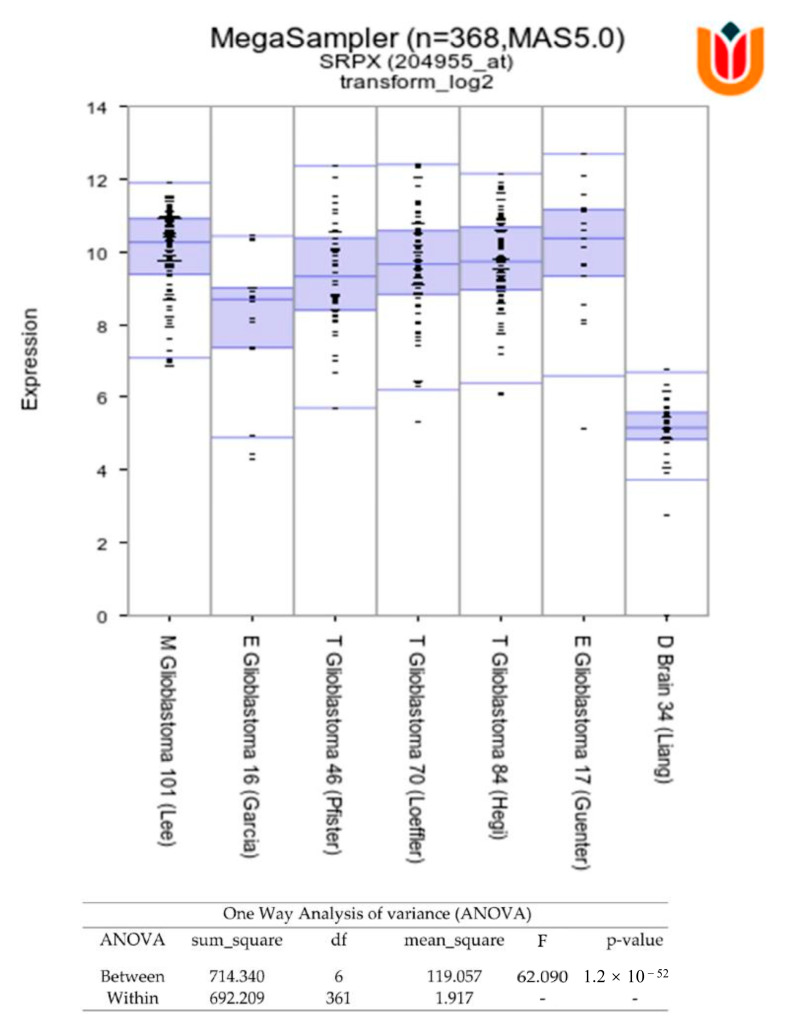
SRPX is overexpressed in glioblastoma tumor tissues. SRPX (ID 8006) expression in glioblastoma tumors (*n* = 334). R2 database was used to perform the analysis.

**Figure 4 cancers-14-01984-f004:**
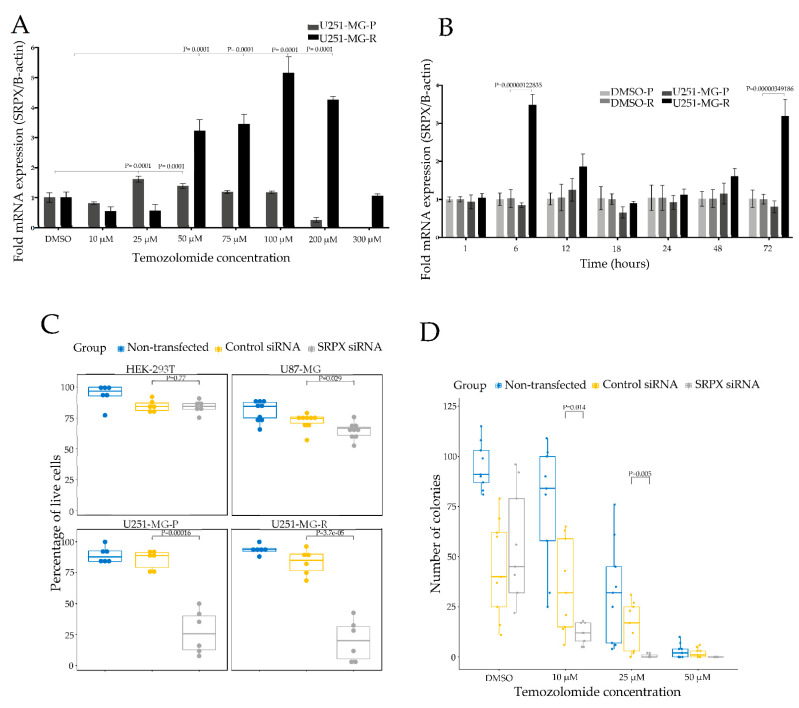
Association of SRPX with TMZ resistance in glioblastomas. (**A**) The mRNA expression levels of SRPX at increasing concentrations of TMZ, and (**B**) at different time intervals. Relative expression analysis was performed using the ΔΔCq method and gene expression was normalized to hB-actin. Error bars indicate means ± SEM and *p*-values indicate significance compared to the DMSO treatment (Bonferroni or Dunnett multiple comparisons test). (**C**) Cell viability. 72 h after transfection, cells were stained with trypan blue and then counted by using a Neubauer cell counting chamber. (**D**) Colony formation assay showing U251-MG-P colony survival after being exposed to TMZ. Error bars indicate means ± SEM and *p*-values indicate significance compared to the cells treated with control siRNA (Bonferroni or Dunnett multiple comparisons test).

## Data Availability

All other relevant data are available from the corresponding author upon reasonable request.

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
