# Peer review of "SRPX Emerges as a Potential Tumor Marker in the Extracellular Vesicles of Glioblastoma"

_cancers, 2022, doi:10.3390/cancers14081984_

Round 1

Reviewer 1 Report

In this very well written manuscript Ampudia-Mesias et al. compared the proteomic profiles of extracellular vesicles (EVs) from primary glioblastoma cell lines and human primary astrocytes, where they identified Sushi Repeat Containing Protein, X-linked (SRPX) as a unique protein expressed in EVs from tumor cell lines. They further validated their findings with functional assays using established glioblastoma cell lines and HEK293T cells, suggesting that SRPX is a putative druggable target, involved in resistance to temozolomide.

While the editing service of this manuscript was exceptional, I would suggest the authors to spell glioma grade with Arabic numerals, as recommended by the 2021 WHO classification of the central nervous system (PMID: 34185076). In addition, while it is implicit that all primary cell lines are IDHwild type, it would be interesting to know the IDH status of the gliomas in the immunohistochemistry (IHC) validation. Because the number of patients for the IHC validation was very low, adding a table with the summary of clinical data (including IDH and MGMT status) of those patients would be very informative.

Furthermore, adding information about the treatment received by the patients with recurrent tumors is important to establish if the behavior of SRPX is the same after treatment with TMZ versus RT+TMZ, which is the standard treatment for glioblastoma.

Finally, other limitations may be discussed for this study. For example, while it is exciting to identify markers from EVs, are there plans to evaluate peripheral blood of patients? This is a key step to move forward the establishment of SRPX as a non-invasive biomarker, highly needed for the follow up of patients with glioblastomas.

Thank you for the opportunity of reviewing this manuscript.

Author Response

We would like to thank the reviewers so much for their constructive comments on the manuscript. We sincerely appreciate reviewers for taking the time to go over the manuscript. The comments have been addressed and helped us to improve the quality of the manuscript.

Reviewer 1:

Comment 1. While the editing service of this manuscript was exceptional, I would suggest the authors to spell glioma grade with Arabic numerals, as recommended by the 2021 WHO classification of the central nervous system (PMID: 34185076).

Author Response: All glioma grades have been updated and labeled according to the 2021 WHO new classification.

Comment 2. In addition, while it is implicit that all primary cell lines are IDH wild type, it would be interesting to know the IDH status of the gliomas in the immunohistochemistry (IHC) validation. Because the number of patients for the IHC validation was very low, adding a table with the summary of clinical data (including IDH and MGMT status) of those patients would be very informative.

Author Response: As requested, we analyzed mutation status of the IDH1, IDH2 genes, TERT promoter and MGMT promoter methylation by pyrosequencing and prepared a table presented as Supplementary Table 2. Accordingly, Grade 2 and 3 samples presented in Figure 2A have IDH1 R132H mutations and WT IDH2 genes. All grade 4 and recurrent samples, IDH1 and IDH2 are wild type and do not have R132H mutations. MGMT promoter was found to be methylated in grade 2, 3, and 4 samples, whereas recurrent tumor of grade 4 showed no methylation in the MGMT promoter. In addition, Figure 2A has been updated.

Comment 3: Furthermore, adding information about the treatment received by the patients with recurrent tumors is important to establish if the behavior of SRPX is the same after treatment with TMZ versus RT+TMZ, which is the standard treatment for glioblastoma.

Author response: The information about treatment received by the patients is enclosed in the Supplementary Table 2.

Comment 4: Finally, other limitations may be discussed for this study. For example, while it is exciting to identify markers from EVs, are there plans to evaluate peripheral blood of patients? This is a key step to move forward the establishment of SRPX as a non-invasive biomarker, highly needed for the follow up of patients with glioblastomas.

Author Response: That is a great point. As we discussed in the original submission that “ …Further studies are required to evaluate and validate in a larger cohort whether SRPX can be a diagnostic, prognostic and/or predictive marker to treatment response, in particular for TMZ in patient serum derived EVs…” We are currently collaborating with Mayo Clinic Rochester, MN USA to obtain approximately 150 glioblastoma pre-surgery serum samples in order to investigate whether SRPX can serve as a diagnostic serum marker in patient serum samples.

Reviewer 2 Report

  • In the current study, the authors aim at identifying biomarkers in Extracellular vesicles that can be collected from the blood and be used as a prognostic and diagnostic marker. Through the current study the authors have shown value in sushi-repeat containing protein X-linked (SRPX) as a highly enriched protein found in EV obtained from GBM cell lines, however, to truly claim this to be a potential diagnostic and prognostic marker the authors should isolate EVs from PBMCs from patients with GBM to confirm this finding. This will greatly support their current findings.
  • In figure 1C the authors claim that “ Strikingly, sushi repeat-containing protein, X-linked (SRPX) was the only protein detected in all glioblastoma-derived EVs and was absent in HPA-derived EVs” however SRPX seems to be at a similar level between HPA EV and IN-GB9 EV. Could the authors clarify this?
  • The authors need to fix the figure number to Figure 2B for: “To validate these observations on larger and independent 242 patient cohorts, we analyzed dataset from the Chinese Glioblastoma Genome Atlas (CGGA)……… found that Grade IV tumor samples had higher SRPX mRNA expression compared to Grade III samples (Figure 2C).”
  • This needs to be corrected to Figure 4B “ but dramatically  dropped to the lower levels at 200 µM and 300 µM TMZ due to cell death (Figure 4-A).”
  • The authors need to rewrite the results for figure 4. The result section is out of sequence.
  • In Line 284 correct U252MG-R cells.
  • The font size for Figures 4C and 4D needs to be increased.
  • In the time-course experiment, the authors show an increase in SRPX mRNA levels at 6 hours in U251-MG-R cells with 200 µM TMZ however there doesn’t seem to be any increase in SRPX mRNA levels at 48 hours. Comparing Figure 4A and 4B, the authors have shown a significantly high SRPX mRNA level in U251-MG-R cells with 200 µM TMZ  when cells were collected at 72 hours. So does SRPX mRNA levels increase at 6 hours and then reduce and increase back at 72 hours? The data seems inconsistent and needs to be clarified.
  • Could the authors comment on why TMZ treatment increases SPRX expression levels in the cell lines in Figures 4A and 4B
  • Figure numbers need to be corrected: “We found that SRPX depletion sensitized U251MG-P to TMZ compared to siRNA-control and untreated cells (Figure 4D). We also observed significantly fewer number of colonies (Supplement Figure 3).”
  • Is there a difference in the expression level of SRPX in U251 cells before and after the cells were made resistant to TMZ? Does silencing SRPX in U251-TMZ resistant cells then sensitize these cells towards TMZ?
  • The human protein atlas dataset shows that SRPX is not prognostic in glioma. The authors also need to discuss and address this.
  • Throughout the manuscript, there are several grammatical and wrongly cited figure numbers that need to be corrected and the manuscript needs to be well proofread.

Author Response

We would like to thank the reviewers so much for their constructive comments on the manuscript. We sincerely appreciate reviewers for taking the time to go over the manuscript. The comments have been addressed and helped us to improve the quality of the manuscript.

Reviewer 2:

Comment 1: In the current study, the authors aim at identifying biomarkers in Extracellular vesicles that can be collected from the blood and be used as a prognostic and diagnostic marker. Through the current study the authors have shown value in sushi-repeat containing protein X-linked (SRPX) as a highly enriched protein found in EV obtained from GBM cell lines, however, to truly claim this to be a potential diagnostic and prognostic marker the authors should isolate EVs from PBMCs from patients with GBM to confirm this finding. This will greatly support their current findings.

Author response: As we discussed in the original submission that “ …Further studies are required to evaluate and validate in a larger cohort whether SRPX can be a diagnostic, prognostic and/or predictive marker to treatment response, in particular for TMZ in patient serum derived EVs…” We are currently collaborating with Mayo Clinic Rochester, MN USA to obtain approximately 150 glioblastoma pre-surgery serum samples in order to investigate whether SRPX can serve as a diagnostic serum marker in patient serum samples.

Comment 2: In figure 1C the authors claim that “ Strikingly, sushi repeat-containing protein, X-linked (SRPX) was the only protein detected in all glioblastoma-derived EVs and was absent in HPA-derived EVs” however SRPX seems to be at a similar level between HPA EV and IN-GB9 EV. Could the authors clarify this?

Author response: This is a good observation, and it needs to be clarified. According to our data SRPX is the only protein enriched in the majority of glioblastoma EVs that was absent in the HPA-derived EVs. The manuscript has been updated accordingly.

Comment 3: The authors need to fix the figure number to Figure 2B for: “To validate these observations on larger and independent 242 patient cohorts, we analyzed dataset from the Chinese Glioblastoma Genome Atlas (CGGA)……… found that Grade IV tumor samples had higher SRPX mRNA expression compared to Grade III samples (Figure 2C).”

Author response: The figure number has been updated.  The corrected number is Figure 2B.

Comment 4: This needs to be corrected to Figure 4B “ but dramatically  dropped to the lower levels at 200 µM and 300 µM TMZ due to cell death (Figure 4-A).”

Author response: As also suggested by other reviewer, Figures 4A and 4B have been switched to reflect the order as they are discussed in the results section. This error has been corrected.

Comment 5: The authors need to rewrite the results for figure 4. The result section is out of sequence.

Author response: We would like to thank the reviewer for this comment. The following section, Association of SRPX with TMZ resistance in glioblastomas, has been rewritten according to the results shown in Figure 4.

Comment 6: in Line 284 correct U252MG-R cells.

Author response: The name of cells has been corrected in the line 284.

Comment 7: The font size for Figures 4C and 4D needs to be increased.

Author response: The font size has been revised.

Comment 8:   In the time-course experiment, the authors show an increase in SRPX mRNA levels at 6 hours in U251-MG-R cells with 200 µM TMZ however there doesn’t seem to be any increase in SRPX mRNA levels at 48 hours. Comparing Figure 4A and 4B, the authors have shown a significantly high SRPX mRNA level in U251-MG-R cells with 200 µM TMZ  when cells were collected at 72 hours. So does SRPX mRNA levels increase at 6 hours and then reduce and increase back at 72 hours? The data seems inconsistent and needs to be clarified.        

Author response: We appreciate this comment giving us an opportunity to clarify our observation regarding the time points of the TMZ exposure. First of all, our three independent experiments consistently showed the same data, namely the upregulation of SRPX mRNA both at 6 h and at 72 h of the TMZ (200 µM) treatment. The Ct values obtained from these experiments are available upon request by the reviewers. We believe that this time-dependent upregulation of SRPX mRNA is most likely associated with the cell cycle arrest caused by TMZ. TMZ is known to cause S/G2M phase arrest, as TMZ interferes with the DNA synthesis through the methylation of nucleobases. It is likely that SRPX is upregulated in the S/G2M phase of the cell cycle upon an exposure to the DNA damaging agent, which is in line with the upregulation of SRPX at the 6 h time point of TMZ treatment. Following TMZ treatment, while many cells die and/or undergo S/G2M cell cycle arrest, some cells exit mitosis and enter the cell cycle, which usually occurs after an elongated repair phase. However, the cells entering the second round of cell cycle that are not fully repaired would not be able to complete the cell cycle on time, but rather would rest in the cell cycle until the cells are able to exit mitosis. It is quite likely that the reason we have seen a peak in SRPX mRNA at 72 h is that our cells entering the second round of the cell cycle were arrested within the cell cycle, and those cells were accumulated at the S/G2M phase at 72 h. Because, as a separate project, we are currently investigating the cell cycle-dependent regulation of SRPX gene regulation both in normal and genomic stress conditions, our upcoming study will address the detailed information on the SRPX transcriptional gene regulation.

Comment 9: Could the authors comment on why TMZ treatment increases SPRX expression levels in the cell lines in Figures 4A and 4B

Author response: The SRPX transcript expression levels are increased in TMZ-treated cells and knocking down of SRPX mRNA via siRNA technology sensitizes glioblastoma cells to TMZ. This data might suggest that SRPX is involved in the TMZ response or resistance.

However, more studies are needed to investigate how exactly SRPX is involved in TMZ resistance in glioblastomas.

Comment 10: Figure numbers need to be corrected: “We found that SRPX depletion sensitized U251MG-P to TMZ compared to siRNA-control and untreated cells (Figure 4D). We also observed a significantly fewer number of colonies (Supplement Figure 3).”

Author response: The number of the figure has been updated, and it is named as Supplementary Figure 5.

Comment 11: Is there a difference in the expression level of SRPX in U251 cells before and after the cells were made resistant to TMZ? Does silencing SRPX in U251-TMZ resistant cells then sensitize these cells towards TMZ?

Author response: In order to address these questions, we performed two experiments as follows.

  1. We performed qRT-PCR from RNA samples isolated from U251-MG-P (Parental cells) and U251-MG-R (TMZ resistant cells) and found that endogenous SRPX mRNA expression levels are significantly higher (~ 2 fold) in U251-MG-R cells. These new data are now presented as Supplementary Figure 3.
  2. We also performed new cell viability experiments using these TMZ resistant cells in the presence of siRNA directed agent SRPX mRNA and found that knocking down of SRPX transcription sensitized U251-MG-R (TMZ resistant cells) to TMZ. These new data are now presented as Supplementary Figure 6.

Comment 12: The human protein atlas dataset shows that SRPX is not prognostic in glioma. The authors also need to discuss and address this.

Author response:  We agreed that the Human Protein Atlas results do not appear to show transcriptional profile changes in SRPX expression in gliomas. However, the information we have uncovered was based on protein expression (Mass Spectral Analysis) which will not be present in the Human Protein Atlas. Our data also suggests that SRPX protein expression will have prognostic value in patients with glioblastoma, as opposed to patients with all grades of gliomas as is analyzed in the Human Protein Atlas dataset.

Comment 13: Throughout the manuscript, there are several grammatical and wrongly cited figure numbers that need to be corrected and the manuscript needs to be well proofread.

Author response:  We apologized for these errors. The manuscript has been corrected accordingly.

Reviewer 3 Report

In this study, Ampudia-Mesias et al analyzed by LCMS four glioblastoma cell lines derived from biopsies and found that the protein SRPX was the most represented. The analysis is qualitative and well explained. however, the Authors entitled the Mat&met paragraph 2.5 "Gel electrophoresis and mass spectroscopy" but the analysis they perform is gel free and no gel electrophoresis is described elsewhere in the manuscript. 

To investigate the role of SRPX in Glioblastoma drug resistance, the Authors generate TMZ resistant U251 MG cell lines. In paragraph 3.2 it is not clear the difference between U251 MG and U251 MG-P cells.

In figure 4 graphs A and B should be inverted since the Authors describe first the results shown in B.

The Discussion should be re-write since it is confusing and completely lacks of consideration about some results.

the title is misleading since it gives the idea that the study is conclusive

Author Response

We would like to thank the reviewers so much for their constructive comments on the manuscript. We sincerely appreciate reviewers for taking the time to go over the manuscript. The comments have been addressed and helped us to improve the quality of the manuscript.

Reviewer 3:

Comment 1: In this study, Ampudia-Mesias et al analyzed by LCMS four glioblastoma cell lines derived from biopsies and found that the protein SRPX was the most represented. The analysis is qualitative and well explained. However, the Authors entitled the Mat&met paragraph 2.5 "Gel electrophoresis and mass spectroscopy" but the analysis they perform is gel free and no gel electrophoresis is described elsewhere in the manuscript. 

Author response: We agree with the reviewer. The mass spectrometry procedure used to analyze proteomics profiles of EVs does not include gel electrophoresis, in this study. Thus, the name of the chapter has been updated to Mass spectrometry.

Comment 2: To investigate the role of SRPX in Glioblastoma drug resistance, the Authors generate TMZ resistant U251 MG cell lines. In paragraph 3.2 it is not clear the difference between U251 MG and U251 MG-P cells.

Author response: We apologized for this confusion, and we clarified the text as follows:  U251-MG-P (parental) and U251-MG-R (TMZ resistant cells).

Comment 3: In figure 4 graphs A and B should be inverted since the Authors describe first the results shown in B.

Author response: These panels have been inverted. Figure 4A and 4B show the dose/dependent and time intervals experiments, respectively. Please see the manuscript.

Comment 4: The discussion should be re-write since it is confusing and completely lacks of consideration about some results.

Author response: As suggested, the discussion has been enriched with more results.

Comment 5: The title is misleading since it gives the idea that the study is conclusive

Author response: As suggested, the title has been revised.

Round 2

Reviewer 3 Report

In this revised version, the Authors have significantly improved the manuscript which is now more intelligible. they responded  all the comments I made. just one point: please, check figure 2 legend (A)

Author Response

Dear Reviewer,

Thank you for your comment on the figure legend 2A, and we apologize for the error.

The figure legend has been corrected accordingly.

Thank you
